# Teaching with Microbes: Lessons from Fermentation during a Pandemic

Megan A. Carney[a]

[a]Anthropology, Center for Regional Food Studies, University of Arizona, Tucson, Arizona, USA

**ABSTRACT** The coronavirus disease 2019 (COVID-19) pandemic introduced unique challenges to teaching at the university level, while also heightening awareness of existing social and health disparities as these shaped interactions and influenced learning outcomes in class settings. Based on ethnographic and autoethnographic data, this article reflects on teaching about human-microbial relations in the context of the course "Anthropology of Food" and specifically at the start of the pandemic. Data demonstrate how students shifted from demystifying microbes to distrusting microbes to reacquainting with microbes through a hands-on experiment with fermentation. The article introduces a microbiopolitical perspective in interpreting students' learning trajectories and ultimate course outcomes.

**IMPORTANCE** As evidenced by classroom experiences in the midst of the COVID-19 pandemic, microbes are "good to teach with" not only within microbiology and related fields but across a variety of academic disciplines. Thinking with microbes is not a neutral process but one shaped by social, political, and economic processes. Imploring students to contemplate how power dynamics and patterns of inequality are detectable at the microbial level may offer a unique opportunity for transforming one's view of the world and our relatedness with both humans and nonhumans.

**KEYWORDS** anthropology of food, biopolitics, fermentation, human microbial relations, microbiopolitics, pandemic, pedagogy, social equity

The coronavirus disease 2019 (COVID-19) pandemic has laid bare the social inequalities that render differential vulnerability to illness and death, including within the realm of higher education (1–3). In addition, the economic, political, and social effects of the pandemic on universities have stimulated more critical attention to university teaching and processes of knowledge production (4–7). Moreover, teaching during the time of COVID-19 has underscored the ways that formal learning environments never neatly adhere to a specific set of temporal and spatial conditions but are only as generative as they are flexible to change.

In this article, in-person and digital ethnographic, as well as autoethnographic, methods inform my reflections on teaching "Anthropology of Food: Culturing Cultures" at the start of the COVID-19 pandemic during the spring 2020 semester at the University of Arizona. In its third iteration, this particular offering of the course engaged with the anthropology of food through the lens of fermented foods and human-microbial relations while integrating theory on settler colonialism, multispecies ethnography, critical food studies, the anthropology of food, science and technology studies, biopolitics, and intersectionality. Although I cannot claim myself an expert on the fermentation process, I nonetheless chose to focus on the social, cultural, and political aspects of fermentation, specifically, its renewed popularity within the domestic sphere (prior to the pandemic), evidenced by the growing number of people, particularly in a North American context, engaged in some form of home brewing, baking, canning, and preserving, as well as on the burgeoning research on the human

Address correspondence to mcarney@arizona.edu.

microbiome, the "probiotic turn," and the mainstream interest in probiotics, prebiotics, and other ostensibly "curative" foods and beverages thought to play a role in contributing to gut health as well as overall physical and mental well-being (8, 9).

Whereas the global-industrial food system promotes homogenization, pasteurization, mass production, processing, and distribution, the revitalization of fermented foods poses a direct challenge to this system (10). The global-industrial food system comprises part of the "antibiotic era," deemed by some as a defining biopolitical logic of the Anthropocene (9). Moreover, the global-industrial food system threatens biodiversity in its quest to eradicate any and all potential "risks," specifically, pathogens and bacteria; paradoxically, these efforts make the system as a whole structurally insecure and more vulnerable to disease (11). However, fermented foods have also come to index technologies of the self; i.e., they comprise part of a self-care regimen in the context of neoliberal governmentality that has increasingly transferred responsibility for health and well-being from the realm of the collective to the individual (9, 12, 13).

In responding to recent calls that anthropologists in particular "capture transforming intimacies and changing discourses of human-microbial life during and after COVID-19 both in the home and in other environments" (14), I hypothesize that microbes are "good to teach with" for their potential to underscore the interrelatedness of all life and, concomitantly, because of the need for cultivating greater intra- and interspecies awareness (15). More than a decade of ethnographic fieldwork in the western United States and in the Mediterranean had led me to contemplate the structural dimensions of uneven microbial exposure and distribution, theorizing microbiopolitics, and engaging with fermentation, especially in my teaching, for its political potential as a site of transformation (12, 16).

Drawing on my training as an anthropologist and ethnographer, my objects of analysis in the pages that follow are the power dynamics in which the above-mentioned course took place, how these dynamics were underscored in our course materials, and the ways that these dynamics mapped onto students' experiences with the course and the pandemic. In particular, this article discusses biopolitics (17–19) and microbiopolitics (10, 20) as they intersected over the span of the semester at three scales: (i) as a theoretical framework for the course materials, specifically, through differential access to fermented foods and exposure to microbes (21, 22); (ii) in the context of the differential effects of the pandemic and how these were refracted in differing interpretations of the course materials; and (iii) within our class and along lines of social differences that translated into differential engagement and participation in the class and, in some cases, forced withdrawal.

## RESULTS

**The first 8 weeks (prepandemic): making friends of microbes.** Mid-January 2020, the course began with readings and other materials that introduced students to the concept of fermentation, i.e., the transformation of one substance by microorganisms, such as bacteria, yeasts, or mold, into another substance that is "more digestible, nutritious, and delicious" (23), as well as to the concept of biodiversity and the accelerated decline of biodiversity in our natural environment, foods, and guts (24). We read Katherine Harmon Courage's book *Cultured* and articles in medical anthropology outlining the connection of food to human health. We also watched the 2017 film *Fermented* to explore the ubiquity of fermented foods and beverages across human cultures, the connection of these foods to livelihoods, and the rich, centuries-old repositories of knowledge that sustain the production and sharing of these foods (25; see also *Fermented*). We discussed how digestion itself is a highly evolved process and how humans have coevolved with microorganisms to use sensory cues, such as taste and smell, to discern what is edible from what is potentially noxious. Course materials emphasized the ways that human labor is both necessary and passive in the process of fermentation, i.e., involving "a vibrant entanglement of the human providing ideal conditions for the microorganisms to thrive" (26). We delved into recent research on the

human microbiome to examine more closely the ways that fermented foods may serve therapeutic purposes and counteract the effects of the "war on germs," i.e., globalized practices of prescribing antibiotics, sanitizing environments, and obsessive application of antibacterial substances that have paralleled the rising global incidence of metabolic illness, inflammatory diseases, respiratory illness, and psychiatric disorders (9, 25, 27). Alongside examinations of how cultures around the world integrate food as the "frontline of healing" (28), we also critiqued profit-making motives of the agrifood industry to commodify and mass distribute probiotics as an ostensible cure-all for unhealthy guts and called for greater attention to how foods affect human microbiomes and overall health in local contexts (i.e., how foods are grown, prepared, and consumed, as well as local biology, ecology, and microbial populations) (29). Students reflected on the food- and herb-based remedies utilized within their households, families, and broader social networks and developed a greater appreciation for understanding that there is no singular "right" way to eat (30–32).

With these themes as an entry point to the course, we engaged Benezra et al., "Anthropology of Microbes," as well as Swanson et al., *Domestication Gone Wild: Politics and Practices of Multispecies Relations*, to situate contemporary debates on the interrelatedness of diet, microbes, and human and environmental health and to analyze fermentation as both a cultural product and a site of human-microbial relations (33, 34). We examined the linkages among domestication, fermentation, and the origins of civilization, for instance the domestication and cultivation of grains for the purposes of making beer (35, 36). We extended our thinking to account for the ways that domestication has contributed to biodiversity loss and the proliferation of plagues, processes that were particularly pronounced in, and instrumental to, settler colonialism. We discussed how European settlers introduced nonnative domesticated animals to the Americas along with pathogens that were fatal for native populations, both human and nonhuman, utilizing multimedia representations of these events, such as Mark Dion's *Conquistadors* (1961), depicting European domesticated animals as the original "*conquistadores*" that wiped out many species of animals native to the Americas. Students began to connect the dots between these historical conditions and the structure of today's global-industrial food system, characterized by industrial monocultures, homogenization of foodstuffs (declining biodiversity of plants and animals), and widespread availability of calorie-dense and nutrient-poor processed foods with uneven consequences for human health and livelihoods.

Our conversations turned more explicitly to concerns about social equity as we welcomed renowned food justice activist Karen Washington as a guest speaker; her visit was scheduled as part of the "Just Nourishment" programming organized by the UA Center for Regional Food Studies and the Tucson-based Dunbar Pavilion, an African American culture and arts center. Washington recounted for us how she had been inspired to convert empty lots in her native Bronx neighborhood into an urban agricultural oasis, while observing the toll of a heroin epidemic and growing rates of asthma on her community. She explained that "Food justice doesn't exist. . .unless you are actively working on injustices," such as reparations for those who were enslaved and displaced by centuries of settler colonialism and, more recently, through gentrification (37–39). She argued for "food apartheid" as a conceptual framing superior to "food deserts" for talking about the struggles of people of color, demonstrating how discourse around food deserts reinforced feelings of alienation among socially and economically marginalized people. She also told us about starting the Black Farmers Fund as a means of building wealth in Black communities and combatting the anti-Blackness of capitalist systems, which had excluded people like her from "a seat at the table"; instead, these systems had objectified and ingested them or, as she stated, "put us on the menu."

Meanwhile, students reported back on current events that related to course themes, often highlighting trends in fermented products, i.e., craft beer, kombucha, and yogurt, and analyzing how these foods were marketed to a particular privileged

class of consumers, or they featured nutrition and dietary interventions designed to address health disparities, noting how interventions often overlooked important factors, such as race, class, and other markers of social difference, as well as structural constraints that narrowed one's dietary options.

As we approached spring break in late February and even into early March, the likelihood of a full-fledged pandemic seemed like a rather distant possibility. While we had briefly discussed media coverage of the situation in China and warnings by officials in the United States about possible disruptions to life here, many of my students were still gearing up for travel over the upcoming university closure. On Thursday, 5 March 2020, we bid our farewells, completely naive to the fact that this would be our last in-person meeting.

**The pandemic breaks spring break: an unwelcome return to "microbes as parasitic foe."**

> I want to write and reassure everyone that this course will proceed (mostly) as planned but with the bulk of our activities occurring online. We are all dealing with extraordinary circumstances, but we will be okay. . . .
> —My correspondence to students on 15 March 2020, 8:12 a.m. MST

> We went into spring break, and everything seemed calm; little did we know that we would not be going back to campus again.
> —Excerpt from a student response paper, April 2020

The World Health Organization declared a pandemic only a few days after students departed campus for spring break. Not only were they informed by university administration that all courses would transition to online, but they were also expected to immediately vacate campus housing. As students responded to myriad disruptions to their everyday lives, they struggled to focus on their studies, as well as acclimate to an entirely new modality of learning. Meanwhile, I was facing struggles of my own at home with regard to balancing work with childcare after my daughters' school closed its doors. Despite my sincere efforts to convey a sense of calm to students in my post-spring break correspondence, none of us were actually "okay." For these reasons, I significantly altered evaluation criteria for the course and, when possible, extended opportunities to students to reflect on what was happening around them.

With the written assignment that followed our transition to online, students were invited to reflect either on recent course materials or about how the pandemic had affected their lives. Most students chose the latter theme, relating experiences of being displaced from university housing and severed from the broader campus community (e.g., "It has been a little difficult to wrap my head around the fact that I was not able to appreciate the little things one more time before I left Tucson"), moving "back home" and dealing with tense familial interactions, losing jobs and having to apply for unemployment (most of them for the first time) and other financial struggles (e.g., "My father owns a small seafood restaurant, and our income is down about 50%. [He is] stressed about losing the business and thousands of dollars. He's put in a request with the government's PPP [Payroll Protection Plan] to access a loan so that our business is able to stay afloat but unfortunately there are thousands of other businesses [doing the same]"), struggling to manage coursework amid Internet connectivity issues or a lack of necessary technology (e.g., "It has been a struggle to gather any remaining motivation to continue to work and even finish my classes"), being estranged from family (especially for international students) and/or being close to older and at-risk relatives, and completely halting social interactions with friends.

They also lamented the cancellation of major life events, such as graduations (e.g., "Although I am glad to be finishing my college career, I am disappointed I will not be able to finish with the entirety of the college experience; this is even more so being a first-generation student") and study abroad plans; one student's spring break study abroad trip to Paris ended abruptly, and her grandmother died of organ failure upon her return to the United States. They revealed fears about graduating and entering the

workforce during a recession (e.g., "Many companies and organizations have altered their hiring status and many decisions have been postponed causing a lot of stress for me") and plans to apply to graduate, law, or medical school being derailed due to cancellations or postponements of upcoming standardized test dates. Some of them shared their experiences as "essential workers" (e.g., "On a daily basis, customers are rude at my job, they yell at us, demand extra services for free, forgetting to realize that WE are there every day risking our lives for THEM") and the shock of coming back to a ghost town after spring break (e.g., "It was almost eerie being back; the life seemed to have vanished from the once-bustling city"). Yet they also alluded to taking better care of themselves and those around them, such as older relatives and neighbors (e.g., "Self-care is often overlooked during the hectic hustle and bustle of our former every-day lives"; "Cooking at home has been a major change in my and my family's life since quarantining"). With consent, students had an opportunity to read and respond to their peers' reflections, an exercise that proved mildly cathartic for some.

These life- and course-related disruptions also overlapped with our transition to the second half of the semester, in which we were to learn directly from Tucson-based fermented food entrepreneurs, including brewers, bakers, coffee roasters, a chocolatier, and a cheese-monger. Unfortunately, most of these individuals were facing significant business challenges of their own in adapting to the conditions of the pandemic, and only two were able to arrange for virtual guest visits with the class.

During this time, students' current event presentations shifted to topics around the effects of COVID-19 on food insecurity and the food system. Examples of story titles selected by students included "How to Safely Feed Food-Insecure Seniors During a Pandemic" (40), "SNAP Could Feed an Economic Recovery During and After the Pandemic" (41), "Farmworkers Are in the Coronavirus Crosshairs" (42), "Food Shortages? Nope, Too Much Food in the Wrong Places" (43), "Critical Food and Farm Rules Have Been Rolled Back Amid Pandemic" (44), "Community Supported Agriculture Is Surging Amid the Pandemic" (45), and "Can Restaurants Survive the Pandemic by Feeding Those in Need?" (46).

In a cruel twist of fate, students expressed ambivalence about their recently acquired open-mindedness toward and knowledge about microbes, including the central role that these play in human health. They resumed "fearful intimacies" in the realm of human-microbial relations (14), despite the coronavirus being a virus and not a microbe; that viruses exist at the scale of microbial communities and can profoundly alter the behavior of living organisms made it difficult to decipher the two. They observed and assiduously adhered to necessarily draconian practices of self-quarantine, sanitation, and the wearing of personal protective equipment required for flattening the curve and controlling the transmission of COVID-19. Meanwhile, the social and economic disparities that had factored centrally in our critical analysis of course materials in the earlier half of the semester were now shaping differential vulnerability to the virus as well as reinforcing patterns of advantage and disadvantage among students, notable for instance in terms of those who were able to continue with their participation in the class versus those whose access was inhibited or compromised in one way or another. There were several students from whom I did not hear after spring break; some of them eventually requested to formally withdraw from the course. Others I heard from toward the end of the semester apologizing for their absence and explaining their lack of access to the appropriate technologies (i.e., laptops or desktop computers, reliable Internet) required for remote learning.

**Sweet (and sour) surrender: the fermentation project.** The final weeks of class were dedicated to students' fermentation projects. Originally planned as an activity for small groups, the pandemic had rendered impossible or unfavorable in-person gatherings among students. As described in the syllabus, this assignment called for students to

> gather in groups of three to four people according to a fermented food of interest; collectively research recipes for the fermented food; collectively seek ingredients for the recipe; prepare and consume the fermented food as a group;

mSystems®

and prepare a presentation. . .in which you reflect on your collective fermenting experience.

I encouraged students to document their experiences with photos and to take notes on sensory observations, i.e., relating to sight, smell, taste, touch, throughout the process. Although groups had formed prior to leaving for spring break, i.e., prior to the pandemic, I sought input from students about how we could still proceed with the assignment amid the unanticipated challenges that we were then all encountering. Students elected to continue with the fermented food that they had chosen and coordinated with other members of their group to prepare the food "together, apart" (a phrase inspired by *The New York Times* podcast "Together. Apart" that launched in April 2020, described as "part guide, part reminder of the resiliency of the human spirit to still creatively meaningfully gather, even while we have to be apart"). Groups formed to make a range of fermented foods, including *dosa* (rice pancake made from fermented batter), sourdough bread, cultured butter, *tepache* (a fermented beverage made from the peel and rind of pineapple), fermented soda, *tempeh*, and *mageu* (a fermented beverage made from maize meal).

As highlighted in their final group presentations on the last days of class, students appreciated the fermentation project for providing a means to stay connected with their peers during the pandemic. At a time of immense physical and social isolation, students connected with their group members, often across vast distances, by sharing updates on their progress with the project through images and written observations. For instance, the group that prepared sourdough bread compared photos of the bubbling surface on their sourdough starters from day 1, 2, etc., and sensory experiences, describing aromas that were "warm, sweet, and tangy." As one member of this group emphasized, "It's fascinating to see the comparison of the different sourdough recipes especially because we're all in different places; two of us are in Arizona, I'm in Canada, and [name of student] is in Tennessee." Students from the *dosa* group planned an imaginary meal together consisting of *dosa*, traditional potato filling, and chutney and divided up preparation of its parts.

Students described challenges in obtaining ingredients and having to adapt recipes amid disruptions to supply chains and hoarding of essential ingredients by panicked consumers. They often resorted to finding creative substitutions, such as using sauerkraut or Greek yogurt as a base when local stores were sold out of starters or yeast. In some cases, they relied on the generosity of neighbors who had supplies to share, e.g., "One of my neighbors just happened to give me some yeast, which I was very thankful for."

For many students, the slow and evolving process of fermentation seemed to catalyze a shift in their relationship to food in which they were not the primary agent of change, e.g.,

> I kept thinking about my [fermented food] and just wanted to check on [the microbes], and like, make sure they were all good. So, I felt I was able to relate to that and have a relationship with food in this process. . . . I feel like I'll probably make my own from now on.

They elaborated on surrendering to the microbes, e.g., "you really just need patience to allow the bacteria to do what it's supposed to do," and on the exhilaration of witnessing microbial transformations.

> It was exciting to see how the bubbles started to come up, signaling that it was alive and the microorganisms were there doing their work. . .and how natural the fermentation process is; you're taking a couple of ingredients and you're letting the ingredients just do their own thing and just letting it sit there.

They also found solace as well as reward and fulfillment in the fermentation process, e.g., "It's really comforting when you're making something that you think is complicated, and everything is actually going according to plan. So, it was nice to do something like that."

The project also indexed for many students how they were experiencing temporal and relational shifts as a result of the pandemic. They described a slowing down in

everyday routines. Students who made *tempeh* for instance had dehulled their soybeans by hand, yet they were unbothered by the amount of time required for such tedious work. One student surmised, "Through this pandemic, we have been making bread just because we have more time, I guess." For some, the foods that they were preparing strengthened their appreciation for family and relatives both living and deceased with whom they shared a connection through food. They enlisted their relatives with whom they were quarantining in the fermentation project and rejoiced in learning how to make foods that were part of their cultural heritage: "It's something that brings me back to my ancestors....this fermentation process [for *tepache*] is something that they have been doing for generations."

In summary, when so many aspects of students' lives seemed unpredictable and out of control, the fermentation project represented something tangible that they could complete at home. Indeed, the pandemic renewed popularity for breadmaking and other do-it-yourself domesticity (26). Although students had to navigate obstacles, such as working together remotely and experimenting with substitutes for unobtainable ingredients, they seemed to derive considerable pleasure from engaging with their peers and others in their lives in an exercise of surrendering to the microbes. The material aspects of sociality that had been so prominent in their prepandemic lives resurfaced to some degree in the context of fermentation, through their interactions both with the microbes and with each other. While giving their final presentations, students acknowledged the ways that the fermentation project afforded them space to process the pandemic and its many material and immaterial implications.

## DISCUSSION

**Microbiopolitical pedagogies.** In developing this course a few years ago and in refining the curriculum over time, building from the intellectual contributions of Michel Foucault, Thomas Lemke, Michelle Murphy, Nikolas Rose, Heather Paxson, and others, while using fermentation as an operational framework, I have subscribed to the notion that a microbiopolitical perspective, i.e., examining biopolitics at the microbial level, holds political potential for redressing inequities in society (20). In other words, imploring students to contemplate how power dynamics and patterns of inequality are detectable at the microbial level may offer a unique opportunity for transforming one's view of the world and our relatedness with both humans and nonhumans. In short, I hypothesized that microbes are "good to teach with," a hypothesis that has since been overwhelmingly supported with evidence not only from the classroom but also in the midst of a pandemic.

Our class served as a microcosm of the broader debates, anxieties, and forms of resistance animating the contemporary moment as one of a global pandemic or as the collision site of biopolitical logics and biosecurity concerns (21). The intellectual engagements with which we concerned ourselves over the first 8 weeks of the semester became real in practice through the pandemic, requiring that I as the instructor and my students as participants in the course attenuate to our collective needs as they surfaced in the moment. This teaching experience provided further evidence that formal learning environments must adapt to shifting circumstances of the moment. Formal learning environments are also relational. Despite our best efforts to control what happens over the span of a semester and prescribe a path of study through a prefixed recipe, i.e., a syllabus, rapidly changing social, political, and environmental conditions require us to cultivate response techniques. Similarly, "fermentation does not take place along a linear trajectory set forth by the human instigator; it does not progress from origin to destination solely because of human intervention. Rather, it is emergent and context-dependent because it is dialogic" (15). As such, our pedagogical logics might benefit by taking cues from fermentation logics. How might fermentation both literally and metaphorically shift possibilities for politics and relationality (specifically our ethical obligations and affective entanglements) with both the human and nonhuman?

As posited by Ishaq et al. and further examined by the others in this special issue, "social inequality, which impedes access to macrobiodiversity, also impedes access to microbiodiversity and the health benefits therein" (47). Risk of infectious disease manifests differentially and in accordance with (micro)biopolitical logics that allow certain lives, including microbial lives, to flourish and others to perish. These inequities in microbial distribution and exposure cannot be remedied through modifications to individual behavior but rather demand swift and comprehensive structural changes.

Regrettably, but also unsurprisingly, "health" continues to be approached as a matter of individual responsibility in the context of COVID-19. Institutions of higher education have been among the leading culprits; as college campuses gradually reopen, for instance, administrators have tried to absolve themselves of responsibility while overlooking how risk maps onto extant forms of social inequality. As I was putting final touches on this paper in the fall of 2020, college campuses had become the newest hot spots for COVID-19 outbreaks. These experiences, no doubt, will shape theories of (micro)biopolitics for years to come and likely inspire more attention to the biopolitics of higher education.

## MATERIALS AND METHODS

Autoethnographic and ethnographic data presented for analysis in this article were gathered *post hoc* from my written and virtual records of the class, including our syllabus, classroom observations, student essays (anonymized and excerpted with the authors' permission), and notes from presentations.

While I did not collect demographic information, most of the 41 students enrolled were in their early 20s, more than half had declared majors in either anthropology or food studies, and many were in-state, first-generation college students, reflecting the broader demographics of the student population at the University of Arizona.

## ACKNOWLEDGMENTS

Many thanks go to Sue Ishaq and Michael Friedman for organizing this collection and to the anonymous reviewers. Additional thanks go to my students from the course "Anthropology of Food," who inspired this article and whose contributions to class continue to challenge my thinking and shape my pedagogy, as well as to members of Nutrire CoLab.

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
