## [Reviewer comments · mSystems]

Teaching with Microbes: Lessons from Fermentation During a Pandemic

Megan Carney

Corresponding Author(s): Megan Carney, University of Arizona

Review Timeline:

Submission Date:	May 6, 2021
Editorial Decision:	May 28, 2021
Revision Received:	June 2, 2021
Accepted:	June 3, 2021

Editor: Suzanne Ishaq

Reviewer(s): The reviewers have opted to remain anonymous.

Transaction Report:

DOI: <https://doi.org/10.1128/mSystems.00566-21>

May 28, 2021

Dr. Megan A Carney
University of Arizona
School of Anthropology
Tucson, AZ 85716

Re: mSystems00566-21 (Teaching with Microbes: Lessons from Fermentation During a Pandemic)

Dear Dr. Megan A Carney:

Thank you for submitting your manuscript to mSystems. We have completed our review and I am pleased to inform you that, in principle, we expect to accept it for publication in mSystems. However, acceptance will not be final until you have adequately addressed the reviewer comments.

The reviewers and I agree that this is a well-written and intriguing piece, which provides interesting perspective on the pandemic and human interactions with microbes. A few minor suggestions have been made to clarify some aspects of the language.

In addition, your contribution has been flagged for needing formatting changes to comply with the guidelines for research articles. In particular, an Abstract, and an Importance, section should be added, and the citation style on your references should comply with the ASM style. You may also need to consult with editorial production staff on whether and how to incorporate additional headings (such as Methods and Results) into your manuscript in order to comply with journal guidelines without disrupting the flow of your narrative, and I can facilitate this.

Thank you for the privilege of reviewing your work. Below you will find instructions from the mSystemseitorial office and comments generated during the review.

Preparing Revision Guidelines

For complete guidelines on revision requirements, please see the Instructions to Authors at <https://msystems.asm.org/sites/default/files/additional-assets/mSys-ITA.pdf>. **Submissions of a**

paper that does not conform to mSystems guidelines will delay acceptance of your manuscript.

Sincerely,

Suzanne Ishaq

Editor, mSystems

Journals Department
Reviewer comments:

Reviewer #1 (Comments for the Author):

This is a creative and pedagogically sophisticated approach to teaching and thinking with microbes. The paper builds on rigorous research and provides insightful ways to bridge the gap between research and teaching. Most importantly, the author ensures the readers understand that thinking with microbes is not a neutral process but one shaped by social, political and economic processes. A really incredible contribution. I can see this paper having a great impact.

Reviewer #2 (Comments for the Author):

This is an outstanding article, with broad relevance for teaching in the pandemic and post-pandemic era. It uses the topic of microbes to shed light on both an effective teaching strategy and broader dynamics of power in the food system and beyond. The concept of "Microbiopolitical Pedagogies" is indeed "good to think with;" readers, whether they are students or professors, will find this article's exploration of how one classroom adjusted to feeding, cooking, and teaching during covid thought-provoking and inspiring. I have a few small comments, which I make in the hope of improving an already impressive text. 1) I think it might be useful to clarify in the first paragraph of the section "the first 8 weeks" that this happened before the pandemic. While careful readers will know this, at first I did not realize that this was documenting pre-pandemic era-teaching. This explanation would also be helpful for readers who are not on the US semester system, who might not be familiar with a typical semester timeline. 2) Also, I wondered if the

pandemic was on students' minds and part of the discussion before the WHO declaration, or if students were largely unaware? (For example, I had a Chinese student in my spring term class who began to live in lockdown in January because she was following international news. Had students already begun to change their lives before the break and might these changes have impacted the class?). As currently written, it seems that the pandemic appeared over spring break, and a bit of foreshadowing what students were thinking in the classroom in the weeks leading up to the change to online education -- and all that was to come -- might be helpful. 3) As a final suggestion, I think it might be useful to tease apart the relations between microbes and viruses slightly more (the paper suggests that students went from being accepting to becoming fearful of microbes. But covid is a virus, not a microbe. Did they understand this distinction? What did they make of it? Those suggestions aside, I learned a lot about effective classroom instruction during covid from this article.

Reviewer #1 (Comments for the Author):

This is a creative and pedagogically sophisticated approach to teaching and thinking with microbes. The paper builds on rigorous research and provides insightful ways to bridge the gap between research and teaching. Most importantly, the author ensures the readers understand that thinking with microbes is not a neutral process but one shaped by social, political and economic processes. A really incredible contribution. I can see this paper having a great impact.

Thank you for this positive feedback.

Reviewer #2 (Comments for the Author):

This is an outstanding article, with broad relevance for teaching in the pandemic and post-pandemic era. It uses the topic of microbes to shed light on both an effective teaching strategy and broader dynamics of power in the food system and beyond. The concept of "Microbiopolitical Pedagogies" is indeed "good to think with;" readers, whether they are students or professors, will find this article's exploration of how one classroom adjusted to feeding, cooking, and teaching during covid thought-provoking and inspiring. I have a few small comments, which I make in the hope of improving an already impressive text. 1) I think it might be useful to clarify in the first paragraph of the section "the first 8 weeks" that this happened before the pandemic. While careful readers will know this, at first I did not realize that this was documenting pre-pandemic era-teaching. This explanation would also be helpful for readers who are not on the US semester system, who might not be familiar with a typical semester timeline.

This is a wonderful suggestion and I added a few words at the start of this section to clarify that it was leading up to the pandemic.

2) Also, I wondered if the pandemic was on students' minds and part of the discussion before the WHO declaration, or if students were largely unaware? (For example, I had a Chinese student in my spring term class who began to live in lockdown in January because she was following international news. Had students already begun to change their lives before the break and might these changes have impacted the class?). As currently written, it seems that the pandemic appeared over spring break, and a bit of foreshadowing what students were thinking in the classroom in the weeks leading up to the change to online education -- and all that was to come -- might be helpful.

Another excellent suggestion. I added a paragraph at the end of this section that addresses these questions.

3) As a final suggestion, I think it might be useful to tease apart the relations between microbes and viruses slightly more (the paper suggests that students went from being accepting to becoming fearful of microbes. But covid is a virus, not a microbe. Did they

understand this distinction? What did they make of it? Those suggestions aside, I learned a lot about effective classroom instruction during covid from this article.

I added a couple of clauses to when first discussing the transition to “microbes as foe” to touch on this important point.

June 3, 2021

Dr. Megan A Carney
University of Arizona
School of Anthropology
Tucson, AZ 85716

Re: mSystems00566-21R1 (Teaching with Microbes: Lessons from Fermentation During a Pandemic)

Dear Dr. Megan A Carney:

Your manuscript has been accepted, and I am forwarding it to the ASM Journals Department for publication. For your reference, ASM Journals' address is given below. Before it can be scheduled for publication, your manuscript will be checked by the mSystems senior production editor, Ellie Ghatineh, to make sure that all elements meet the technical requirements for publication. She will contact you if anything needs to be revised before copyediting and production can begin. Otherwise, you will be notified when your proofs are ready to be viewed.

We recognize that the video files can become quite large, and so to avoid quality loss ASM suggests sending the video file via <https://www.wetransfer.com/>. When you have a final version of

the video and the still ready to share, please send it to Ellie Ghatineh at eghatineh@asmusa.org.

Sincerely,

Suzanne Ishaq
Editor, mSystems

Journals Department
Phone: 1-202-942-9338